



# Workflow for Systematic Design of Electrochemical In Operando NMR Cells by Matching $B_0$ and $B_1$ Field Simulations with Experiments

Michael Schatz[1,2], Matthias Streun[3], Sven Jovanovic[1], Rüdiger-A. Eichel[1,4], and Josef Granwehr[1,2]

[1]Institute of Energy Technologies, Fundamental Electrochemistry (IET-1), Forschungszentrum Jülich, Jülich, Germany
[2]Institute of Technical and Macromolecular Chemistry, RWTH Aachen University, Aachen, Germany
[3]Central Institute of Engineering and Analytics, Electronic Systems (ZEA-2), Forschungszentrum Jülich, Jülich, Germany
[4]Institute of Physical Chemistry, RWTH Aachen University, Aachen, Germany

**Correspondence:** Michael Schatz (m.schatz@fz-juelich.de)

**Abstract.** Combining electrochemistry (EC) and nuclear magnetic resonance (NMR) techniques has evolved from a challenging concept to an adaptable and versatile method for battery and electrolysis research. Continuous advancements in NMR hardware have fostered improved homogeneity of static magnetic field, $B_0$, and radio frequency field, $B_1$, yet fundamental challenges caused by introducing essential conductive components into the NMR sensitive volume remain. Cell designs in EC-NMR have largely been improved empirically, at times supported by magnetic field simulations. To propel systematic improvements of cell concepts, a workflow for a qualitative and semi-quantitative description of both $B_0$ and $B_1$ distortions is provided in this study. Three-dimensional Finite Element Method (FEM) simulations of both $B_0$ and $B_1$ fields were employed to investigate cell structures with electrodes oriented perpendicular to $B_0$, which allow realistic EC-NMR measurements for battery as well as electrolysis applications. Particular attention is paid to field distributions in the immediate vicinity of electrodes, which is of prime interest for electrochemical processes. Using a cell with a small void outside the electrochemical active region, the relevance of design details and bubble formation is demonstrated. Moreover, $B_1$ amplifications in coin cells provide an explanation for unexpectedly high sensitivity in previous EC-NMR studies, implying the potential for selective excitation of spins close to electrode surfaces. The correlation of this amplification effect with coin geometry is described by empirical expressions. The simulations were validated experimentally utilising frequency encoded [1]H profile imaging and chemical shift imaging of [1]H, [13]C, and [23]Na resonances of $NaHCO_3$ electrolyte. Finally, the theoretical and experimental results are distilled into design guidelines for EC-NMR cells.

## 1 Introduction

As a versatile tool for non-invasive and quantitative identification of molecules and their chemical environment, Nuclear Magnetic Resonance (NMR) spectroscopy presents a advantageous partner to electrochemistry (EC) techniques. The synergy between these methods enables a comprehensive study of processes and reactions within an operating electrochemical cell, especially when EC methods struggle with ambiguity in the presence of side reactions (Lozeman et al., 2020). Particularly, the integration of magnetic resonance imaging (MRI) and relaxometry has yielded valuable insights into the chemistry inside





operational electrochemical cells (Krachkovskiy et al., 2020; Jovanovic et al., 2023). Unlike optical methods that often focus on electrode surfaces, NMR spectroscopy enables simultaneous investigations of both the electrode surface and the electrolyte composition (Borzutzki and Brunklaus, 2017).

However, in comparison to other spectroelectrochemical methodologies the full potential of EC-NMR techniques has not been fully exploited. Integrating electrodes, wires, and electronic components within the NMR sensitive volume is challenging, as this leads to distortions in the static magnetic field $B_0$ and the radio-frequency (RF) field $B_1$, critical for precise NMR measurements (Pietrzak et al., 2021). Therefore, *in operando* NMR is often a tug-of-war between the intrinsic conductive components of an electrochemical cell and the desire for high field homogeneity, which affect NMR signal sensitivity, resolution, and the quantitativity of the method.

Numerous EC-NMR studies have tackled these challenges in the field of battery research due to the involvement of NMR-active nuclei in redox reactions crucial for battery electrochemistry (Krachkovskiy et al., 2020; Pecher et al., 2017; Hu et al., 2016). A wide variety of cell concepts have been tailored to fit the basic requirements of EC-NMR (Chandrashekar et al., 2012; Bhattacharyya et al., 2010; Krachkovskiy et al., 2016; Kayser et al., 2018). Tang et al. (2021) provide an extensive review of MRI techniques applied to such battery cells. However, the nature and magnitude of $B_0$ and $B_1$ field distortions remained understudied.

To minimise the effects of these distortions, significant effort has been directed towards well-conceived cell designs. Basic requirements are that metallic parts should be non-magnetic and should not encase the cell. Otherwise, RF irradiation is shielded from the sample depending on the thickness and conductivity of the metal, described by the skin effect (Britton, 2014). At the same time, electrical contact has to be ensured, the cell has to be completely sealed to prevent leakage, and the filling factor should be sufficient (Tang et al., 2021). Shielding due to the skin effect can be minimised by avoiding RF penetration through conductive components in order to excite the sample. Still, there are two main types of distortions that cannot be entirely inhibited. First, magnetic susceptibility gradients between metals and their surrounding materials result in inhomogeneous $B_0$. Secondly, eddy currents are induced in metals by RF fields, leading to a spatially and temporally dependent alteration of the $B_1$ field strength. This modulation leads to a loss in quantitativity and altering of the circuit tuning (Britton, 2014; Vashaee et al., 2015; Mohammadi and Jerschow, 2019).

Despite the hurdles posed by these $B_0$ and $B_1$ field distortions, several EC-NMR cell designs were presented over the last decades (Mohammadi and Jerschow, 2019; Pietrzak et al., 2021). The pioneering work of Richards and Evans (1975), where wires just outside the sensitive volume in a standard 5 mm NMR tube were deployed as electrodes, was the basis for studies several decades later, such as the work of Silva et al. (2019). They omitted sample rotation, as NMR hardware has improved in terms of of $B_0$ homogeneity, and mixing by external pumping was replaced by making use of the magnetohydrodynamic effect at high magnetic field strength. This effect arises from the Lorentz force acting on ions in solution, resulting in a convective flow that resembles an internal mixer in an *in operando* NMR cell (Benders et al., 2020).

Several other types of cells were developed that aimed at mitigating field distortions by minimising metal content in the cell, either by thin metallic films or foils (Webster, 2004; Zhang and Zwanziger, 2011; Jovanovic et al., 2021; Schatz et al., 2023), electrodes placed outside the sensitive volume (Hallberg et al., 2008), or non-metallic electrodes of low conductivity



(Klod et al., 2009; Bussy et al., 2013). Despite these advancements, the quantitative prediction of $B_0$ and $B_1$ field distortions remained challenging. Understanding the nature and magnitude of these distortions is critical, as it facilitates quantitative data evaluation or tailoring of pulse sequences, e.g. via techniques such as optimal control.

In the few selective studies found in literature on the topic of understanding and mitigating field distortions in EC-NMR, the main focus is typically the skin effect and $B_1$ distortions. Britton (2014) examined the influence of electrode orientation on the $B_1$ field, employing [1]H MRI to study the electrolyte surrounding a metal strip. Minimised distortions were achieved by parallel alignment of the metal relative to the orientation of RF irradiation (Mohammadi and Jerschow, 2019). Jovanovic et al. (2021) confirmed this using nutation experiments with varying angle between electrode and $B_1$ field. To comprehend the skin effect in a Li electrode at varying $B_1$ field directions, Ilott et al. (2014) conducted comprehensive calculations and validated their findings utilising [7]Li MRI. Their investigations expanded into the influence of $B_0$ field variations on the chemical shift of Li metal, arising from susceptibility effects and Knight shift. Vashaee et al. (2015) employed SPRITE pulse sequences that are largely immune to susceptibility and gradient effects. By linearly increasing the flip angles, the signal intensity increases with $B_1^2$, which is utilised to determine the spatial distribution of $B_1$ precisely (Vashaee et al., 2013). This allowed for distinct investigation of the eddy current effect of RF excitation near metal strips, emulating electrodes used in electrochemical applications. Their experimental findings were validated by $B_1$ field simulations in electrode proximity. Numerous combinations of relative $B_0$ and $B_1$ field orientation, imaging pulse sequences as well as cell geometries using flat electrodes were tested by Serša and Mikac (2018) and compared in terms of signal intensity. Eddy currents were identified as the main reason for signal loss. Thus, a parallel orientation of $B_1$ to the electrode was found to be optimal, while the influence of $B_0$ field orientation was minor. Purely phase encoded single point imaging showed the best performance.

Zhang and Zwanziger (2011) employed this to advantage by implementing a parallel-plate resonator in an imaging probe for the investigation of flat membranes in a fuel cell setup. By maximising the filling factor and minimising $B_1$ field distortions, high-resolution imaging could be applied to thin films despite the presence of auxiliary conductive components. Simulations by Walder et al. (2021) showed that between electrodes placed parallelly to $B_1$ no significant decrease of RF field was expected, even inside a metal casing typical for industrial coin cells. Subsequently, they demonstrated first [7]Li and [19]F *in operando* NMR measurements of a commercial coin cell considering the calculated modulations of $B_0$ and $B_1$ field.

While these studies have significantly contributed to comprehending RF field variations, there persists a need for workflows that include $B_0$ and $B_1$ fields simulations of realistic cell setups alongside with experimental validation. With the predominant focus on simulations and experimental determination of the $B_1$ field, the significance of $B_0$ fields often goes overlooked. Moreover, the utilisation of higher magnetic fields, beyond 2.4 T that have been applied in previous studies, remains largely unexplored in this context.

In this study, a workflow that simultaneously considers $B_0$ and $B_1$ field distortions within an electrochemical NMR cell is presented. This approach involves simulating these fields using Finite Element Methods (FEM) and subsequently validating the findings experimentally through imaging techniques. Adapting the optimised EC-NMR cell for $CO_2$ electroreduction from our previous studies (Schatz et al., 2023), eddy currents were reduced to a minimum. Utilising the $B_0$ gradients provided by a diffusion probe, imaging along the axis perpendicular to the electrode was applied to determine local variations of $B_0$ and



$B_1$ field along this specific dimension. Due to the symmetry of the cell setup, the latter was the most relevant dimension of the setup to probe spatial variations of $^1$H, $^{13}$C or $^{23}$Na resonances. This allowed not only to test whether $B_1$ field distortions can really be considered minimal, but also revealed even signal enhancing effects. Regarding $B_0$, field mapping using chemical shift imaging (CSI) of various nuclei is introduced.

## 2 Experimental

A reference sample was prepared following the design of the electrochemical cell presented in our previous publication (Schatz et al., 2023) that was aligned with the optimal cell designs outlined in existing literature: The Cu working electrode (WE) was thin, oriented in parallel with $B_1$, and perpendicularly to the $B_0$ field and $B_0$ field gradients. Reference electrode, counter electrode, the contacting wire and cell holder of the described setup were omitted. $^{13}$C-enriched 1 M NaHCO$_3$ in aqueous solution was used as electrolyte to enable $^1$H, $^{13}$C and $^{23}$Na MRI. Figure 1.a-b illustrates this sample, depicting two scenarios: one with the notch below the electrode filled with electrolyte and the other with air. The purpose of this notch in the electrolysis cell setup was facile contacting of the electrode using a Cu wire and optional mounting of the electrode inside it. The two scenarios were selected for investigation as electrolysis experiments were conducted *in operando* with air underneath the electrode. The air-filled notch could moreover be considered as a model for gas entrapment in the cell housing or gas bubbles forming during electrolysis.

A Bruker DiffBB BBO broadband diffusion probe, which offers magnetic field gradients along the $z$-axis (direction of $B_0$, as shown in figure 1.a), was operated on a Bruker Avance III HD spectrometer (Bruker BioSpin GmbH, Rheinstetten, Germany) with a 14.1 T wide-bore magnet. To distinguish between effects of cell and holder geometry on the $B_0$ field and the influence of the electrode, shimming on the cell was performed without the Cu foil thoroughly. Instead of aiming for the minimum linewidth, the spatial homogeneity of $B_0$ over the $z$-dimension was priorised during shimming and validated afterwards by $^{23}$Na CSI. $^{23}$Na linewidths of ca. 8 Hz and $^1$H linewidths of ca. 6 Hz could be achieved. The Cu electrode was subsequently added to the cell, either with air or with electrolyte in the cylindrical notch underneath. Without further adjustments of the magnetic field, purely phase encoded CSI was performed, as described previously (Schatz et al., 2023). The field of view was set to 20 mm. With 128 points in the spatial dimension and zero-filling to 256 points, the spatial resolution was 78 μm. Here, the $^1$H resonance of water as well as the $^{13}$C and $^{23}$Na resonance of the NaHCO$_3$ electrolyte were imaged along the $z$-axis. Comparable local changes in chemical shift across all three nuclei would indicate a nucleus independent chemical shift (NICS), implying purely a magnetic field effect. Additionally, a frequency encoded $^1$H profiling pulse sequence, also described previously (Schatz et al., 2023), was performed. With 664 points in a field of view of 20 mm, the spatial resolution was 30 μm. Here, a flip angle below 90° was employed, such that a locally increased $B_1$ field would result in increased intensity.

To validate the observations, FEM simulations of the setup shown in figure 1 were conducted using the magnetostatic simulator of the software FEMM (Version 4.2, David Meeker). For the electrode, material properties of copper were chosen, and for the surrounding liquid pure water was assumed. The magnetic susceptibility of PEEK was set to $-9.335\cdot10^{-6}$ (Wapler et al., 2014). Before the simulation the boundary conditions of a volume with cylindrical geometry were optimised in order





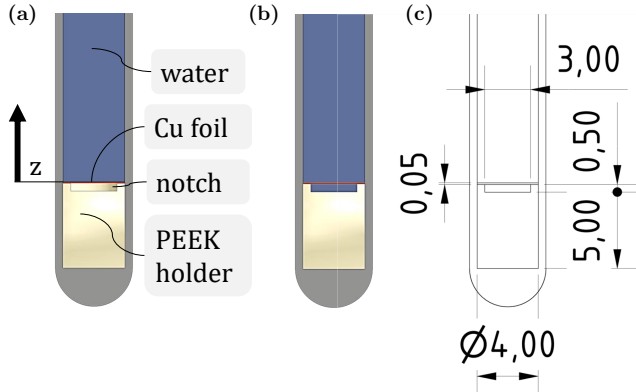

**Figure 1.** (a) – (b) Sectional side view of experimental setup for investigations of changes in the $B_0$ field without and with water in the notch underneath the Cu WE, respectively. (c) Technical drawing with exact dimensions given.

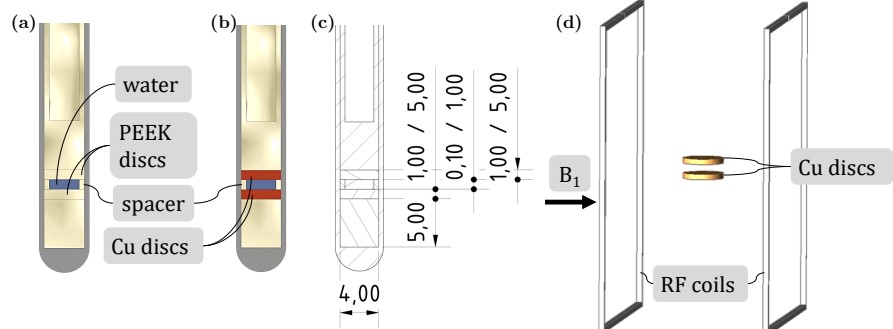

**Figure 2.** (a) – (b) Sectional view of the experimental coin cell setup for investigations of changes in the $B_1$ field with PEEK and Cu discs, respectively. (c) Technical drawing of the sectional view with exact dimensions given. (d) Setup in FEM simulations using EMpro showing the RF coils and the resulting $B_1$ field direction parallel to the discs.

to obtain a uniform magnetic field. This corresponds to shimming of the magnet in the experiment and could be done for an empty volume but also for a volume that is already populated with parts of the setup, e.g. PEEK holder and liquid without the electrode. Next, simulations were performed for the setup with and without the Cu electrode. The change of $B_0$ was obtained as the ratio of the calculated field distributions.

To investigate the $B_1$ field variations in a coin cell for battery applications, a sample imitating such a cell was prepared as shown in figure 2.a-c. The thickness of the Cu discs was 1 mm or 5 mm and the distance between the discs was 0.1 mm or 1 mm. All four possible combinations of disc thickness and distance were tested. To fix the distance, ring spacers made of PEEK were utilised, cf. figure 2. The volume between the discs was filled with high performance liquid chromatography water (Sigma Aldrich Chemie GmbH, Taufkirchen, Germany). To exclude water from the sensitive volume above and below the coin

cell, solid PEEK cylinders were used to fill this space.





Variation of the $B_1$ field between the coins was determined by nutation experiments. The evolution of the water resonance intensity was investigated while the pulse length was linearly increased at constant pulse power. First, the nutation frequency was determined for a sample without conductive metal discs. Instead, PEEK coins of the same dimension were introduced. This is subsequently compared to the nutation frequency with Cu coins in the sample.

The distortion of the $B_1$ field surrounding the metal electrodes was calculated by numerical simulation using EMpro (Version 2020, Keysight Technologies). To mimic a uniform RF field close to the electrodes, a square Helmholtz coil was included in the simulation, comprising two parallel square-shaped wires spaced at 0.5445 times the length of each side of the square. The coil material was assumed to be an ideal conductor, and both square sections of the coil were concurrently driven by a current source. Two discs representing the coin cell were placed centrally between the Helmholtz coils. The dimensions of the discs 145 and the distance between them were varied according to the variations in the experiment, while the material properties of the coins were either that of PEEK or Cu. As the wavelength of RF fields in high-field NMR are on the order of metres, while geometries in EC-NMR cells are on the orders of millimetres, it was assumed that magnetostatic laws can be applied, thus simulations were performed time-independently (Walder et al., 2021).

Analogously to the FEM simulations described above using EMpro, a copper disc, with identical dimensions to the Cu 150 electrode in figure 1 (4 mm diameter, 50 μm thickness) was positioned centrally between the two Helmholtz coils. A local increase in $B_1$ field intensity results in an increased flip angle (Hoult and Richards, 1976). Thus, such a locally increased $B_1$ field was detectable as peak in frequency encoded $^1$H profiles if excitation was conducted using a flip angle of less than 90° for the bulk electrolyte.

## 3 Results and Discussion

Compared to an NMR tube filled only with liquid electrolyte, the insertion of the PEEK holder already introduces a nucleus independent degradation of NMR spectra due to susceptibility differences between liquid and polymer causing $B_0$ distortions. Therefore, the $B_0$ field was homogenised by shimming after insertion of the PEEK holder, but before introduction of any conductive material. As a result, a high spatial homogeneity of the NMR spectra with maximum 0.05 ppm deviation of the peak position and line widths of 8 Hz for $^{23}$Na could be achieved. The corresponding $^1$H, $^{13}$C and $^{23}$Na CSI spectra after 160 shimming are depicted for $z = [0, 2]$ mm in figure 3.a-c, respectively, which are used as a reference for all further experiments.

Afterwards, the Cu foil was inserted while maintaining the position of the cell relative to the centre of the sensitive NMR volume. As a result, local changes in the chemical shifts for all three nuclei were observed, which are depicted in figure 3.e-g for the case of air and 3.i-k for the case of electrolyte in the notch underneath the Cu foil. In general, peak splitting and shifting of the same magnitude on the ppm scale were observed for $^1$H, $^{13}$C and $^{23}$Na measurements, indicating that this effect 165 is nucleus independent and can be attributed to magnetic field effects. Interestingly, significant differences in the extent of magnetic field alterations were observed between the two scenarios of air and electrolyte underneath the Cu WE. In the case of an air-filled notch, the resonance developed both a new, ca. 3 ppm downfield shifted and a ca. 1 ppm upfield shifted feature in electrode proximity. With increasing distance to the electrode, the two features converged, resulting in the bulk resonances





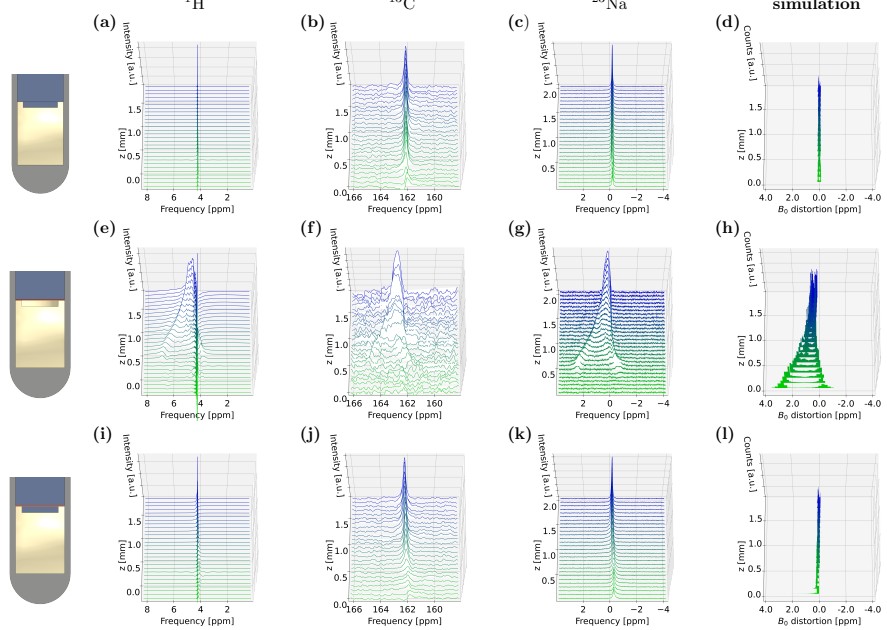

**Figure 3.** Spatially resolved NMR spectra of CSI measurements of experimental setups (a) – (c) without Cu foil, (e) – (g) with Cu foil and air underneath and (i) – (k) with Cu foil and electrolyte underneath, depicted between $z = [0, 2]$ mm. Spectra are shown for $^1$H, $^{13}$C and $^{23}$Na, respectively. Simulated $B_0$ distortions, represented by histograms over the same $z$ interval, for the case of (d) no Cu foil and water filled notch, (h) air and (l) water underneath the Cu foil, respectively. The histograms were corrected under the assumption of perfect shimming of the sample without Cu foil.

that is significantly broadened, even in 2 mm distance to the Cu foil. In the case of an electrolyte-filled notch, no peak splitting

was observed. Instead, only the single peak was shifted to lower frequencies by ca. 0.5 ppm. In both scenarios and for all nuclei, overall decreased signal intensities were observed in spectra corresponding to volumes close to the Cu foil, especially for $z = [0, 0.5]$ mm. This is expected in CSI measurements near paramagnetic electrons on electrode surfaces with additional distortions by gradient pulses and echo formation (Bazak et al., 2020). In quantitative measurements, these disruption of CSI images can be corrected by calibration of the intensity profile using pulse sequences with small angle excitation and stabilised

gradients (Sethurajan et al., 2015; Krachkovskiy et al., 2016; Serša and Mikac, 2018; Bazak et al., 2020). Lastly, the positioning of the Cu foil with respect to the $z$-axis and the insertion of the tube were not perfectly reproducible and, therefore, have to be considered as potential error sources.

To compare the simulation results to these spatially resolved NMR spectra, distributions of $B_0$ variation from excitation frequency, depicted as histograms, were calculated for all voxels in slices of 100 µm thickness in $z$-direction. The simulations

for $z = [0, 2]$ mm are shown in figure 3.h and 3.l with air and water underneath the electrode, respectively. A reference calculation without Cu foil and with a water filled notch was used to correct these histograms assuming a perfect shim on the sample without conductive material. The weighted average of each histogram of this simulation was subtracted from the respective his-





togram representing the same slices. The results of all three simulation scenarios were corrected accordingly and are depicted in figure 3.d. Uncorrected histograms are given in figure A1 of the appendix. The simulated frequency distributions are in good

agreement with the experimental CSI spectra. For the air-filled notch, a splitting of ca. 4 ppm for the two spectral features was observed, which decreases to zero within 2 mm along the $z$-axis. The upfield peak predicted by simulations was shifted by ca. 3 ppm, while the downfield peak was shifted by ca. 1 ppm, which is in quantitative accordance with experiments. For the case of electrolyte under the electrode, the simulation showed no significant difference to the reference simulation without Cu. Thus, the simulation results reflected experimental findings both qualitatively and quantitatively, albeit a post-processing

correction had to be applied assuming a perfect shim. Even though uncorrected, the $B_0$ distortions, depicted in figure A1, show good qualitative agreement with experimental data. Here, the calculated values of the distorted signals were shifted upfield compared to experimental results, which could be attributed to an unshimmed magnetic field. Shimming could only be carried out by post-simulation data processing, and the minimised discrepancy between experimental results and corrected simulations validates this processing step.

Overall, the agreement of simulation and experiment demonstrate that CSI is capable of probing the influence of conductive cell components on $B_0$, and that FEM calculations are suitable to predict the spatially resolved magnetic field distortions in the proximity of conductive materials and their effect on NMR spectra.

The results from combined experiments and simulations indicated that a conductive electrode in the NMR sensitive volume will induce a systematic error in imaging and spectroscopy experiments. By careful shimming after inserting the full cell setup

including all electrodes and wires, this error may be reduced. However, it can not be ruled out that a chemical shift alteration near the electrode remains. Before every *in operando* measurement, the change of chemical shift that remains after shimming has to be determined and, subsequently, corrected. In our previous study, the peak splitting illustrated in figure 3.e-g could be homogenised to a single peak with linewidth on the order of ca. 6 Hz by shimming. A chemical shift difference between near-electrode and bulk electrolyte remained even after long equilibration time and thorough shimming and had to be corrected

during post-processing (Schatz et al., 2023). In particular, local differences to the bulk chemical shift in the reference CSI measurements were subtracted from chemical shift profiles acquired during electrolysis.

The greater impact of the air pocket compared with an electrolyte-filled notch highlights not only the necessity to minimise entrapped air in EC-NMR cell setups and reduce the number of different materials with high susceptibility differences, but also to mitigate the evolution of gas bubbles during electrolysis. While the inhomogeneity caused by air pockets can mostly be

corrected by shimming before the electrolysis, the gas bubble distortions can not be shimmed during electrolysis as they build up, grow and potentially detach in irregular patterns. If gas bubble formation cannot be inhibited, a robust shim is required and acquisition time has to be sufficiently long to average out errors over time. Moreover, multi-nuclear measurements can be utilised to correct for nucleus independent chemical shift effects of gas bubbles during post-processing.

The total intensity of $^1$H density from frequency encoded images is plotted over the $z$-axis in figure 4.a for the experimental

setup with an air-filled notch. Here, a pulse length of 16.5 µs was used, while the pulse duration for a 90° magnetisation flip was determined to be 18.5 µs. Thus, the increased intensity close to the electrode at $z = 0$ mm is attributed to a locally enhanced $B_1$ field. Towards positive $z$-values, the $^1$H density approaches zero at $z = 15$ mm, delineating the upper boundary of the sensitive





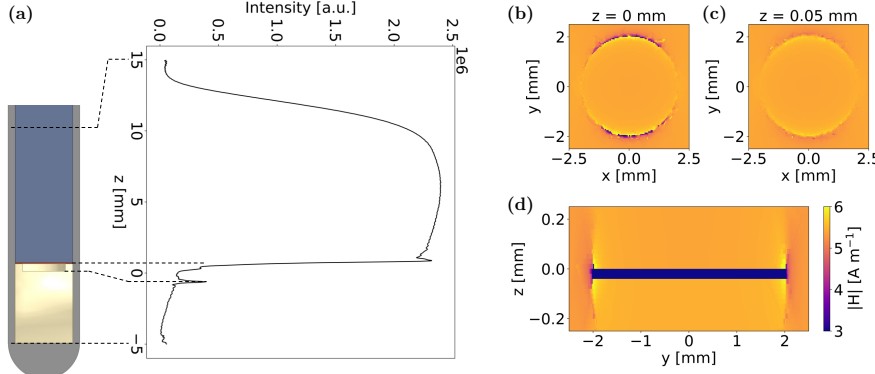

**Figure 4.** (a) Frequency encoded $^1$H density profile of the experimental setup with horizontally placed Cu foil. (b) – (c) $B_1$ field intensities in layers exactly on top of the Cu disc and 50 μm above it, respectively. (d) $B_1$ field intensities in the sectional side view through the Cu disc.

volume. In the negative $z$-range, low $^1$H density indicates the position of the PEEK holder that does not yield a comparably high $^1$H signal intensity as the liquid electrolyte. Residual $^1$H signal and the peak at ca. $z = -1$ mm result from thin films of electrolyte on the surface of the PEEK holder.

Figure 4.b-d presents the outcome of three-dimensional $B_1$ field simulations utilising EMpro. The resulting $B_1$ field intensities are illustrated in top view on the Cu disc, precisely atop and 50 μm above its surface, alongside a sectional view through the disc's centre. Notably, the simulations confirm a localised increase in $B_1$ field near the WE by about 20 % compared to the bulk induced by eddy currents, primarily evident at the edges of the Cu foil. Concurrently, $B_1$ field distortions extend less than 50 μm into the $z$-direction, predominantly confined to volumes close to the electrode edges. Depending on the exact cell geometry and setup, this effect requires consideration in quantitative experiment evaluations. However, for the cell setup utilised in this work, the associated error from $B_1$ field distortions is estimated to be relatively minor. In recent imaging experiments with purely phase encoded pulse sequences, the spatial resolution was on the order of 100 μm and could therefore not resolve this effect.

In case of a second conductive disc added to the $B_1$ field simulations to mimic a coin cell, a sectional view through the centre of the discs is depicted in figure 5 for 1 mm and 0.1 mm distance and 5 mm and 1 mm coin thickness, respectively. Notably, a uniform increase in $B_1$ field intensity was observed between the coins in all cell configurations. With increasing coin thickness and decreasing distance between coins, this effect is further amplified. Further examination of the $B_1$ field along the $x/y$-direction parallel to the disc surface on a line precisely in the centre between the two discs, as depicted in figure 6.a for the four aforementioned combinations of coin thickness and distance, quantified the two overarching trends: an increase in $B_1$ field with 1) decreasing distance and 2) increasing thickness of discs. The percentage increase in $B_1$ intensity between the electrodes compared to bulk values, as determined from simulations, is contrasted to the experimentally measured increase in nutation frequency in table 1. The experimental value was determined by assessing the nutation frequency of the water resonance between non-conductive PEEK discs and conductive Cu discs, cf. figure 2.a and 2.b, respectively. Details about the





determination of nutation frequencies are given in appendix B. Given that the nutation frequency is directly proportional to $B_1$ field, the accordance between simulated and experimental data substantiates that coin cells in parallel orientation to the $B_1$ field exhibit a significant $B_1$ amplification for the volume in between. Compared with other studies calculating $B_1$ intensity between parallel plates (Zhang and Zwanziger, 2011; Walder et al., 2021), where no RF field intensification was observed, setups with larger electrode thickness and smaller distance were investigated, which may not correspond to dimensions of commercial coin

cells, but helped to uncover the amplifying effect and enabled its measurement by nutation experiments.

To extrapolate the extent of this effect, additional calculations were conducted with the coin cell arrangements that led to maximum $B_1$ amplification, i.e. with coin thickness of 5 mm and a minimum distance of 0.05 mm, where the distance is on the order of realistic coin cell dimensions. While the extreme value of either coin thickness or distance was kept constant, the respective other value was varied in the interval [0.05, 0.1, 0.5, 1, 2, 3, 4, 5] mm. The RF amplification is displayed as a

function of the varied value for the two extreme cases in figure 6.b. The $B_1$ profile between the coins in parallel direction to the coin surface is displayed in figure B3 in the appendix. The cell configuration with 0.05 mm thickness and distance that is best comparable to commercial coin cells exhibited an RF amplification of less than 5 %, which could easily be overlooked or interpreted to be within error margins when measuring nutation frequency in a coin cell for *in operando* NMR investigations (Kayser et al., 2018; Walder et al., 2021). Even though electrode thicknesses in the order of 1 mm are not realistic for

commercial cells, this effect could be utilised for *in operando* NMR cells with amplification of $B_1$ field of 90 % and more. With increasing coin thickness or distance, the amplification effect increases or decreases exponentially, respectively, evolving towards a limit. Exponential fits for the two simulation series are plotted as dashed lines in figure 6.b, described by

$$a(\delta) = 74.2\,\% \cdot e^{-0.472\,\mathrm{mm}^{-1} \cdot \delta}, \quad R^2 = 0.976, \tag{1}$$

with RF amplification $a$ in % and coin distance $\delta$ at constant thickness of 5 mm, and

$$a(d) = 77.5\,\% \cdot (1 - e^{-0.707\,\mathrm{mm}^{-1} \cdot d}), \quad R^2 = 0.997, \tag{2}$$

with coin thickness $d$ at constant distance of 0.05 mm, respectively. From these two correlations, a unified equation could be found,

$$a(\delta, d) = 77.9\,\% \cdot e^{-0.472\,\mathrm{mm}^{-1} \cdot \delta} \cdot (1 - e^{-0.707\,\mathrm{mm}^{-1} \cdot d}), \tag{3}$$

which is derived in appendix C.

Eddy currents that are induced in the conductive material perpendicularly to the $B_1$ direction cause a circulating current along the coin surface . The currents on the surfaces of the two coins facing each other cause modulations of the $B_1$ field, but in contrast to the scenario with just one coin, these surface currents are aligned in the same direction and therefore cause spatially uniform amplification of $B_1$. Resonance effects between the coins comparable to a waveguide effect could potentially be accountable for the RF amplifying phenomenon, though further investigation via time-resolved simulations would be required

to elucidate the underlying processes definitively. Regardless of the exact mechanism taking place, the closer the amplifying currents are located to each other, i.e. with decreasing coin distance, the greater the intensification of RF field. Increased





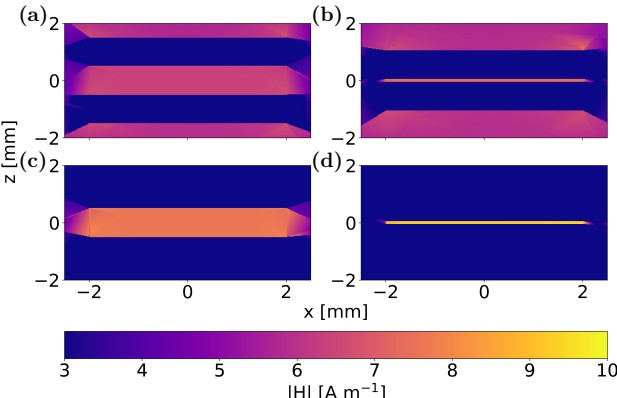

**Figure 5.** Spatial distribution of calculated $B_1$ field in a central sectional plane perpendicular to the coin surface with coins of (a) 1 mm distance and 1 mm thickness, (b) 0.1 mm distance and 1 mm thickness, (c) 1 mm distance and 5 mm thickness, (d) 0.1 mm distance and 5 mm thickness. The origin of the $z$-axis is centred between the two coins.

**Table 1.** Increase of $B_1$ field in simulations and experimentally determined increase of nutation frequency with conductive Cu discs compared to PEEK discs of 1 mm and 5 mm thickness and 1 mm and 0.1 mm distance, respectively.

| Distance | Thickness | Increase of $B_1$ field in simulation | Increase of experimental nutation frequency |
|----------|-----------|---------------------------------------|---------------------------------------------|
| 1 mm | 1 mm | 19.1 % | 14.3 % |
| 1 mm | 5 mm | 37.6 % | 53.9 % |
| 0.1 mm | 1 mm | 41.7 % | 61.6 % |
| 0.1 mm | 5 mm | 74.3 % | 92.4 % |

thickness of the conductive discs has two effects, further increasing eddy currents along the surfaces facing each other. First, with increasing thickness the overall eddy current strength is enhanced proportionally to the interface of conductive material perpendicular to $B_1$. Secondly, in a thicker coin the opposing surface currents are further apart, which decreases attenuation between them.

An additional simulation of $B_1$ field distribution of the coin cell setup is depicted in figure D1 of the appendix, in which the coins were directly employed as part of the resonant circuit. Here, the excitation pattern were fundamentally different from excitation using an external coil, which provides an approach to explain the non-intuitive correlation of $B_1$ amplification with coin thickness. The $B_1$ distribution was strongly influenced by eddy currents at the edges of the coin, which created homogeneous $B_1$ fields inside the coin cell structure. However, when excitation occurs directly through the coins, different eddy currents occur and this homogeneity collapses, and instead, highly localised $B_1$ amplifications occur. In contrast to external excitation, coin thickness variations did not lead to significant differences in $B_1$ distributions.





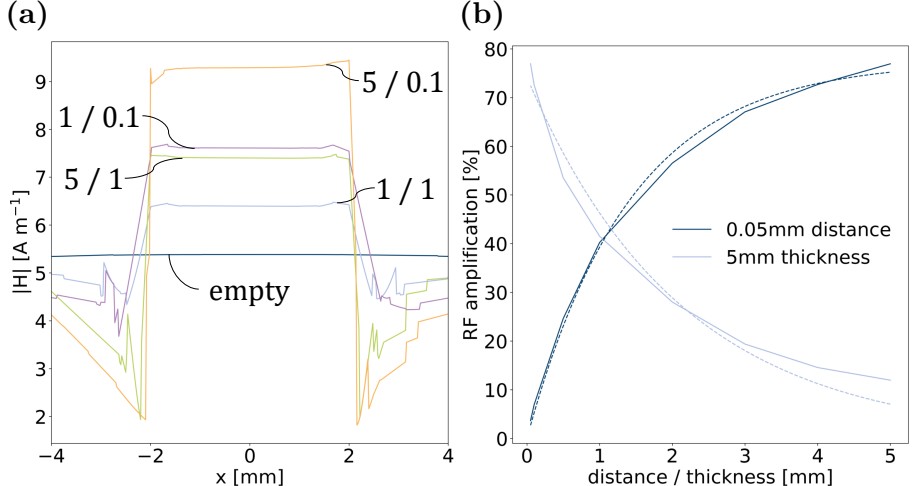

**Figure 6.** (a) Calculated $B_1$ field from simulations in a line along the $x$-axis, parallel to and central between the coin surfaces. Annotations are in the format "X / Y" with coin thickness X in mm and coin distance Y in mm. The calculated $B_1$ field in an empty volume without conductive coins is plotted as reference. (b) RF amplification in comparison to $B_1$ field in an empty volume for two extreme conditions: For a coin distance of 0.05 mm, thickness was varied between [0.05, 0.1, 0.5, 1, 2, 3, 4, 5] mm. For a coin thickness of 5 mm, distance was varied between [0.05, 0.1, 0.5, 1, 2, 3, 4, 5] mm. Dashed lines represent exponential fits to the data.

## 4 Conclusions

This study compared predictive simulation methods and their experimental validation for assessing $B_0$ and $B_1$ field distortions
induced by electrically conductive components within the NMR sensitive volume with emphasis of EC-NMR application. FEM simulations highlighted significant impact of materials with magnetic susceptibility differences on $B_0$ homogeneity and demonstrated semi-quantitative agreement with experimental $B_0$ field distortions measured by [1]H, [13]C and [23]Na CSI. It was observed that the presence of an air pocket beneath the electrode posed challenges due to a large difference in magnetic susceptibility compared to the surrounding. Although this inhomogeneity was shown to be largely correctable by careful shimming,
improved susceptibility-matching of cell components could facilitate the $B_0$ field homogenisation. Not only air pockets in the cell setup but also gas bubbles emerging during electrolysis operation should therefore be minimised or discharged from the sensitive volume.

     FEM simulations of $B_1$ field effects were in accordance with local increases in frequency encoded [1]H profiles. Between electrodes of coin cells, a uniformly enhanced $B_1$ field was predicted by simulations and experimentally confirmed by nutation
experiments. The intensification of RF field intensity was found to increase with decreasing electrode distance and increasing electrode thickness. This finding contributes to understanding why EC-NMR battery cells in previous studies (Kayser et al., 2018; Walder et al., 2021) have exhibited performance beyond expectations and how coin cells could be employed to advantage in EC-NMR applications. As the $B_1$ amplification is comparable to waveguide effects, further time-resolved investigations are suggested to understand the nature of this phenomenon. In this context, it is also important to note that the sample itself





becomes an integral part of the resonant circuit of the probe. Once resonance structures are integrated into the NMR sensitive volume, focusing solely on the skin effect is not sufficient and will not yield quantitative results.

Future studies with higher spatial resolution or surface sensitivity should consider locally inhomogeneous nutation frequency. This effect could even be employed to advantage by slice selection inspired by the DANTE sequence (Maffei et al., 1991) or by numerical pulses optimised by optimal control. Such tailored pulse sequences may mitigate magnetic field distortions, or

even enable the selective excitation of certain volumes or species otherwise not possible. As volumes close to the electrode are affected by increased $B_1$ field, this method could differentiate between signals of bulk and near-surface electrolyte. Additional distorting effects caused by gradient switching during diffusion or imaging experiment should likewise be considered in that regard.

Ultimately, all the design guidelines for EC-NMR cells given in this study and in literature can not be comprised to a uni-

versally valid recipe, but should be consulted in their context. Depending on the specific requirements of the research question, design adjustments could be guided towards either high sensitivity, by employing amplification effects or increased filling factor, or high field homogeneity, by susceptibility matching materials and sophisticated placement of conductive components. By utilising the presented workflow of matching $B_0$ and $B_1$ field simulation with imaging experiment, the quality of outcomes of *in operando* EC-NMR experiments can be aligned with the research objectives.

*Code and data availability.* TopSpin raw data of the presented measurements, as well as the results of $B_0$ and $B_1$ field simulations, are available on the repository Jülich DATA at https://doi.org/10.26165/JUELICH-DATA/KJTAXZ. The simulation codes used in this work and all other data are available from the authors upon request.

## Appendix A: $B_0$ field simulations without shim correction

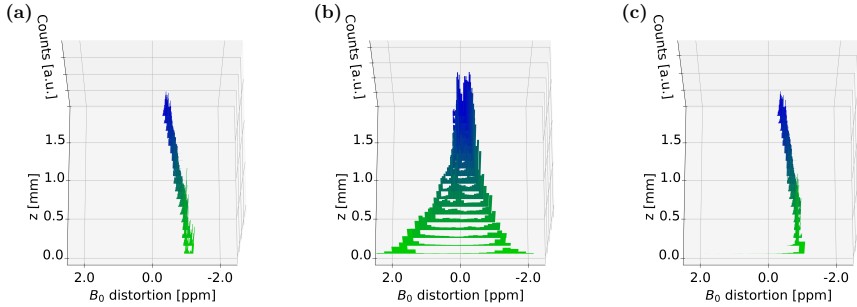

**Figure A1.** Simulated $B_0$ distortions, represented by histograms over the $z$-axis, for the case of (a) no Cu foil and electrolyte-filled notch, (b) air and (c) electrolyte underneath the Cu foil, respectively. The histograms were not corrected in terms of shimming.



## Appendix B: Nutation experiments

The nutation experiments were conducted with increasing pulse length in 80 steps between 5 µs and 400 µs. The delay between two experiments was 5 s, acquisition time was 0.68 s. Spectra were processed with manual phase correction individually for every experiment, line broadening of 5 Hz and no baseline correction. Signal integrals were plotted over pulse length to generate nutation curves. Integral intervals were chosen to fit the resonance frequency of the water between the coins. To distinguish between these resonances and signals originating from the water films between PEEK cylinders and NMR tube glass or in

between PEEK parts, [1]H CSI was employed providing the integral intervals in table B1. Nutation spectra were calculated by Fast Fourier Transform of the nutation signal and plotted in magnitude mode over positive frequency values. Resulting nutation frequencies are displayed in table B1. Nutation curves and spectra of experiments with PEEK and Cu discs of 1 mm and 5 mm thickness and 1 mm and 0.1 mm distance are illustrated in figure B1 and B2.

**Table B1.** Integral intervals and nutation frequency determined for nutation experiments with PEEK and Cu discs of 1 mm and 5 mm thickness and 1 mm and 0.1 mm distance, respectively.

| Distance | Thickness | Disc material | Integral interval [ppm] | Nutation frequency [kHz] |
|---|---|---|---|---|
| 1 mm | 1 mm | PEEK | [4.09, 4.62] | 17.72 |
| 1 mm | 1 mm | Cu | [3.50, 4.84] | 20.25 |
| 1 mm | 5 mm | PEEK | [1.0, 1.21] | 16.45 |
| 1 mm | 5 mm | Cu | [0.82, 1.23] | 25.32 |
| 0.1 mm | 1 mm | PEEK | [3.8, 4.1] | 16.45 |
| 0.1 mm | 1 mm | Cu | [4.1, 4.25] | 26.58 |
| 0.1 mm | 5 mm | PEEK | [4.2, 4.35] | 16.45 |
| 0.1 mm | 5 mm | Cu | [4.0, 4.15] | 31.65 |





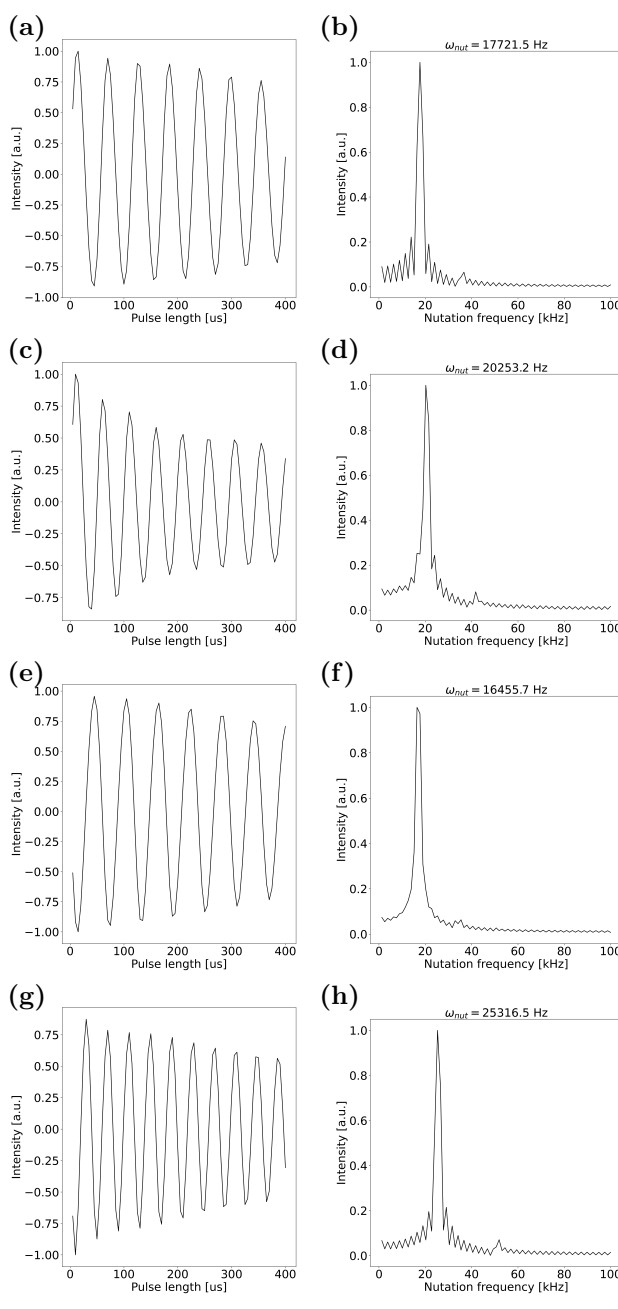

**Figure B1.** (a), (c), (e), (g) Nutation curves and (b), (d), (f), (h) nutation spectra of the $^1$H water resonance in nutation experiments with coin cell setups with 1 mm electrode distance, respectively. (a) – (b) PEEK discs with 1 mm thickness. (c) – (d) Cu discs with 1 mm thickness. (e) – (f) PEEK discs with 5 mm thickness. (g) – (h) Cu discs with 5 mm thickness.

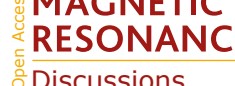



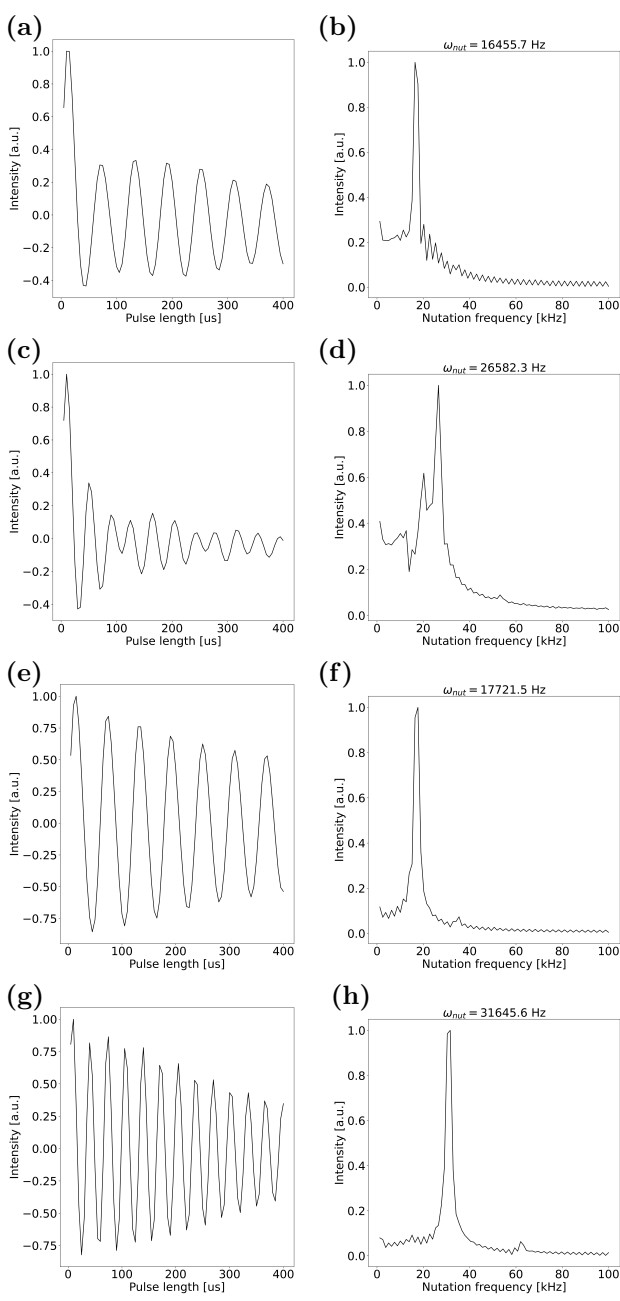

**Figure B2.** (a), (c), (e), (g) Nutation curves and (b), (d), (f), (h) nutation spectra of the $^1$H water resonance in nutation experiments with coin cell setups with 0.1 mm electrode distance, respectively. (a) – (b) PEEK discs with 1 mm thickness. (c) – (d) Cu discs with 1 mm thickness. (e) – (f) PEEK discs with 5 mm thickness. (g) – (h) Cu discs with 5 mm thickness.





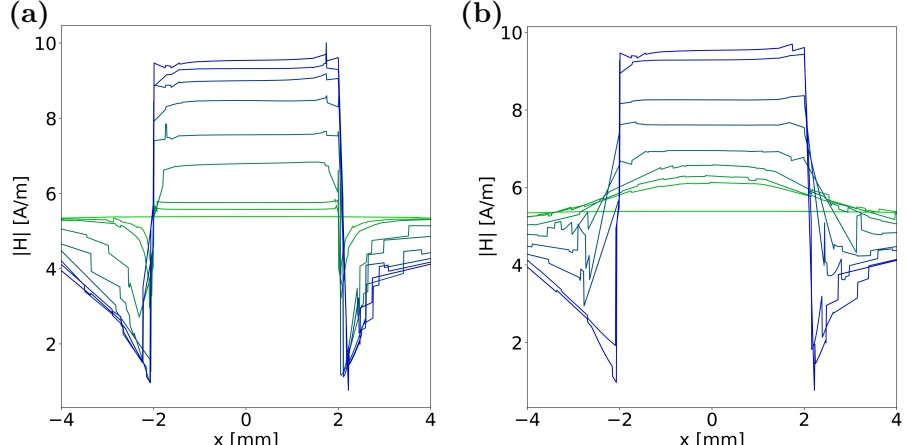

**Figure B3.** Calculated $B_1$ field from simulations in a central line along the $x$-axis, parallel to the coin surface. (a) For a coin distance of 0.05 mm, thickness was increased in the order [0.05, 0.1, 0.5, 1, 2, 3, 4, 5] mm, indicated with changing colour from green to blue. (b) For a coin thickness of 5 mm, distance was decreased in the order [5, 4, 3, 2, 1, 0.5, 0.1, 0.05] mm, indicated with changing colour from green to blue.

**Appendix C: Derivation of a universal empirical equation for $B_1$ field amplification in dependance of coin thickness and distance**

It is assumed that there exists a universal equation that equals equations 1 and 2 and is described by,

$$a(\delta, d) = K \cdot e^{k_\delta \cdot \delta} \cdot (1 - e^{k_d \cdot d}), \tag{C1}$$

with $k_\delta = -0.472\,\text{mm}^{-1}$ and $k_d = -0.707\,\text{mm}^{-1}$ from equations 1 and 2, respectively, and a constant $K$ that has to be determined. Equation C1 has to match equation 1 for coin thickness $d = 5\,\text{mm}$,

$$a(\delta, d = 5\,\text{mm}) = K \cdot e^{k_\delta \cdot \delta} \cdot (1 - e^{k_d \cdot 5\,\text{mm}}) \overset{!}{=} 74.2\,\% \cdot e^{k_\delta \cdot \delta}, \qquad \Rightarrow K = \frac{74.2\,\%}{0.971} = 76.4\,\%. \tag{C2}$$

Same holds for equation 2 and coin distance of $\delta = 0.05\,\text{mm}$,

$$a(\delta = 0.05\,\text{mm}, d) = K \cdot e^{k_\delta \cdot 0.05\,\text{mm}} \cdot (1 - e^{k_d \cdot d}) \overset{!}{=} 77.5\,\% \cdot (1 - e^{k_d \cdot d}), \qquad \Rightarrow K = \frac{77.5\,\%}{0.977} = 79.4\,\%. \tag{C3}$$

As the two calculated values for the constant $K$ differ only slightly, the average of these is used to obtain equation 3,

$$a(\delta, d) = 77.9\,\% \cdot e^{-0.472\,\text{mm}^{-1} \cdot \delta} \cdot (1 - e^{-0.707\,\text{mm}^{-1} \cdot d}). \tag{C4}$$





### 340   **Appendix D: Simulation of $B_1$ field using coin cell electrodes for direct RF excitation**

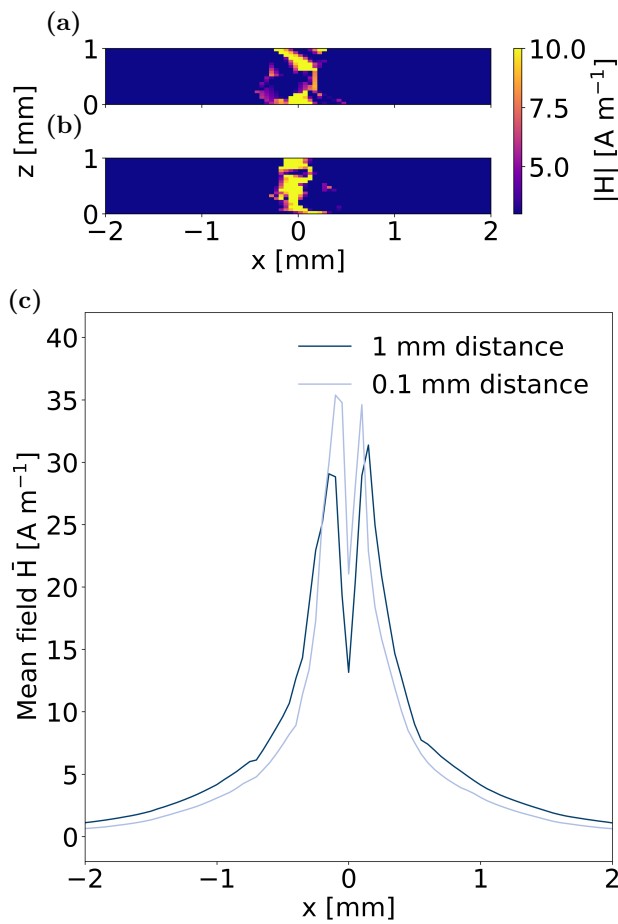

**Figure D1.** Calculated $B_1$ field from simulations using the electrodes of coin cells as RF coil. The magnitude of $B_1$ field in the $xz$-plane between coins with 1 mm distance is depicted for electrode thickness of (a) 1 mm and (b) 0.1 mm. (c) Mean field averaged over $z$-axis plotted over $x$-axis for electrode thickness of 1 mm and 0.1 mm.

*Author contributions.* Each author contributed to this work as follows. MiS prepared the samples and performed the experiments in consultation with MaS, SJ, and JG. $B_0$ and $B_1$ field simulations were performed by MaS. Data analysis and interpretation were performed by MS in collaboration with MaS, SJ, RAE, and JG. The paper was written by MiS in collaboration with MaS, SJ, RAE, and JG. All the authors have read and agreed to the paper.

*Competing interests.* The authors declare that they have no conflict of interest.



*Acknowledgements.* The authors thank Christian Hellenbrandt for support in sample preparation. This research has been supported by the German Research Foundation (DFG) under Germany's Excellence Strategy–Cluster of Excellence 2186 "The Fuel Science Center" (grant no. 390919832).



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
