# Peer review of "Workflow for Systematic Design of Electrochemical In Operando NMR Cells by Matching $B_0$ and $B_1$ Field Simulations with Experiments"

_Magnetic Resonance, 2024_

## Referee Comment (RC3)

The paper by Schatz et al addresses a very important and interesting topic related to MR/MRI studies of electrochemical cells. On the surface the necessity of a strong static field, Bo, switched magnetic fields Gx, Gy and Gz, in addition to a radio frequency excitation field B1, would seem to be problematic in a device that must contain significant conductive and often metallic structures namely electrodes and current collectors for the electrochemical device and ancillary electrical connections. Nevertheless, remarkable progress has been made with MR/MRI studies of electrochemical cells.

Schatz and coworkers have done a good job of assembling and describing the issues. MR/MRI practioners are very enthusiastic about in operando studies but traditional battery researchers and industry are often resistant, since the batteries employed for in operando MR/MRI studies usually don't look like realistic and familiar batteries.

Resolution of this issue in the context of EM compatibility (Bo, Gx,y,z and B1) is very important for wider acceptance of MR/MRI battery in operando studies.

The authors have done a good job of assembling and assessing the pertinent literature with I fear one significant exception. On page 3 the authors assign to Zhang and Zwanziger (2011) the idea of a parallel plate resonator (PPR) for RF excitation in fuel cells. The Zhang and Zwanziger paper is however about the design of an MR compatible electrochemical cell in a 5 mm NMR tube. It is not about a parallel plate resonator, nor a fuel cell. I think the authors may have been intending to reference another work with the same last name of the first author, also in 2011, Zhang and Balcom "Magnetic Resonance Imaging", in 'PEM Fuel Cell Diagnostic Tools', Eds. Wang, H., Yuan, X. and Li, H., Taylor and Francis, Oxford, UK (2011) 229-254.

This paper summarized work by Zhang related to use of the PPR in nafion based fuel cells. The work summarized came from two JMR papers in 2008, one from J Power source 2011 and Can J Chem 2011. The later paper explicitly describes and analyzes the PPR.

The PPR resonator was designed to satisfy the EM compatibility issues outlined in the Schatz paper. I will address this further below but first I will mention that the fuel cell studies of 2008 to 2011 were more recently rejuvenated for studies of lithium ion batteries. There have been a number of published studies, but I will highlight two in particular, (i) Aguilera JMR 2021 (cover photo for the issue). This paper employed extensive use of simulation to design the PPR and critically introduced a cartridge like removable cell. The cell has electrodes that are parallel to the plates of the PPR with all of the attendant EM compatibility benefits. The cartridge is removable permitting multiplexing of samples and the 7Li battery cartridge looks like a battery to traditional battery people. (ii) Goward and co workers have continued to use the PPR resonator and battery cartridge idea in a series of

studies, notably Sanders et al Carbon (2022) wherein they showed both imaging and spectroscopic studies of a lithium ion battery. The PPR / battery cartridge was proven to be a sensitive implementation in the Sanders paper because of the significant sample volume, which is implicit in the resonator design.

But now why does the PPR / cartridge idea work and how does it help EM compatibility? It is easy to explain in words. These reasons are interwoven in the papers mentioned, but the simplest summary is probably the Zhang 2011 chapter. RF shielding must be avoided so that means B1 parallel to the electrodes and other conductive structures as Schatz and coworkers describe. If the electrochemical cell is a capacitor-like design (as in the 7Li cartridge and nafion fuel cell studies) with parallel conductive surfaces (electrodes), there are two possible orientations for Bo (parallel and perpendicular to the plates). The Bo direction will be z of course. It is advantageous for the Bo field, in terms of minimizing susceptibility distortion of the static field experienced by the sample, for the Bo field to be oriented parallel to the conducting surfaces. Okay there are two ways to be parallel to the plates but one of them must be the direction of the B1 field. It must be perpendicular to Bo and parallel to the plates. Let's identify this B1 direction as x. The Bo field, z, once again is parallel to the conductive surfaces of the battery and RF probe. In this geometry then the electrolyte will span the space between the electrodes which will be y.

There is a more subtle but very important consideration for why this geometry is advantageous. It is well known but not always fully appreciated that the Gx, Gy and Gz gradients *all* have z directed fields that vary in the chosen direction (Gx is dBo/dx with Bo z directed). Eddy currents will be at their worst when a z directed field impinges, perpendicular, to a conductive surface and gradients are switched. The emf driving eddy currents results from the time rate of change of flux *through the conducting surface*. The worst case for eddy currents will be a z directed field impinging on a planar conductor that is transverse. The PPR / cartridge combination with the conductors parallel to Bo have minimal conducting cross section in the Bo direction meaning that the emf induced is minimized, minimizing eddy currents. This is true regardless of the direction in which one wishes to do imaging. The above reasoning does not require detailed simulation, it is simple EM considerations, but it is backed up by simulation.

Schatz and coworkers employ electrodes that are perpendicular to the Bo field with gradients that are z directed. This is required given the initial geometry of their cell, magnet and gradient coil. That does not mean they can't achieve good results, they do achieve good results. It does however mean the underlying geometry is non-ideal from an EM compatibility point of view.

The above line of reasoning would be advantageous to include or at least to reference in the paper. It is easy for people to get lost in results and simulation.

On a significant but less serious note, the authors at various times state that SPI, SPRITE, constant time, and chemical shift imaging profile approaches to imaging are advantageous. The implementation that is not explicitly stated, that is similar, is spin echo SPI. The first three methods are FID based. The later two are based on spin echoes. They similar because spatial encoding is purely by phase encode gradients. This is advantageous because if there are susceptibility induced Bo distortions, pure phase encoding will not lead to geometric distortion. The distorted Bo field instead results in a local image contrast (T2* or T2) which may be removed or controlled through choice of imaging parameters. The pure phase encode approach is also robust to image distortion due to eddy currents. With frequency encode imaging eddy currents distort the image geometry. An inhomogeneous static field will also lead to image distortion in frequency encode imaging. These two effects can make it difficult to discriminate B1 inhomogeneity effects, from Bo inhomogeneity effects, from eddy current effects with frequency encode imaging. Eddy currents manifest in pure phase encode image as a change in the image field of view, but not geometric distortion of the object geometry.

Bruce J. Balcom

MRI Research Centre, Department of Physics

University of New Brunswick

---

## Author Response (AR2)

Reviewer 1:

The manuscript presents a good amount of data describing the influence of B0 and B1 distortions in the presence of metal (electrodes) with and without air bubbles underneath. The work describes both experiments and calculations. The main conclusion is that calculations match experiments reasonably well, and that the rf field dependence has a somewhat nonintuitive behaviour. While the work is very important, as a reader, I find it very difficult to follow. The Figures are not very clear and do not highlight very well what needs to be paid attention to. For example, in Fig. 3, it would probably be better to represent the results as 2D contour or pcolor plots, rather than 3D projections. Each panel could be labelled in addition with a descriptive text, which would make it much easier to appreciate what it is showing.

> Thank you for your valuable feedback. We worked on the clarity of all figures and implemented your suggestions, but also applied some further modifications. As the type of spatially resolved spectra, as presented in Fig. 3, is not often published and, therefore, there is no universal way of visualisation of these data, we have discussed various plot options and found the 3D waterfall plots the most expressive. Since both reviewers found Fig. 3 difficult to follow, we agreed to change the style of representation into pseudocolor plots. Even though the narrow lines in some experiments, such as (a) and (i), are now more difficult to recognise, the enhanced visibility of spectral intensity speaks for this representation style. To increase readability of these plots, enlarged sections were included in these subfigures.

Fig. 4 appears to have an odd combination: proton density and rf field distribution, and it is unclear why these two different data sets have been put together.

> The correlation of these values is described in the main text for the chosen pulse length. We added the explanation to the figure caption. Additionally, the figure was rearranged to enhance clarity of the figure.

Fig. 5 could probably benefit from an additional histogram, or some other representation that would allow to identify field changes better.

> An additional plot with a histogram of B1 field distributions was added to the figure. Additionally, the position of Cu coins was marked by dashed lines in the original figure.

Figs. 6 and B3 are probably the most interesting ones, but are very hard to follow, it is not clear in which order the lines were plotted, it seems labels are missing.

> It seems that labels were not shown by the reviewer's pdf reader, because the lines in the plot were labelled directly in the figure without a legend. Here, we noticed that another font type was used for the labels, which has been changed in all figures to the font of the "(a)" and "(b)" labels. However, in this case, a legend was added in Fig. 6a.

So overall I would recommend to enhance the clarity of figures and subsequently update the surrounding explanatory text to help the reader navigate the manuscript.

Reviewer 2:

The manuscript describes a qualitative validation method for evaluating B0 and B1 distortions in NMR techniques used in electric cell research. The study includes two experimental setups that simulate the effects of different components of an electric cell (copper electrode, air bubble) on B0 and B1, respectively. The simulation and experimental results corroborate each other, providing a valuable

reference for studies that combine electrochemistry and nuclear magnetic resonance. The manuscript is well written, and the experiments are carefully designed. For this reason, it is suitable to be published in Magnetic Resonance.

However, there are still several points that require further clarification prior to publication:

1. The manuscript aims to verify B0 and B1 distortions introduced by electric cells in NMR experiments, so the liquid medium should ideally be an electrolyte. However, the authors only used an electrolyte in the first B0 distortion verification experiment. In all other simulations and experiments, water and HPLC water were used as the liquid medium, without any explanation for this change. This should be clarified.

> Thank you for your feedback and thorough reading. This explanation is indeed missing and was added to the experimental section (l. 134f).

2. Page 5, Figure 1: The manuscript indicates that the B0 field direction is aligned with the marked z-axis. It would be helpful to include the B0 field direction in the figure for clarity.

> The direction of B0 field was added to the figure accordingly.

3. Page 6, Line 142: The phrase "0.5445 times the length" is unclear. Is this an empirical value or is it derived from literature? If it is based on literature, a reference should be provided.

> This is indeed not an arbitrarily chosen value, the reference was added accordingly.

4. Page 7, Figure 3: The figure is well-designed, but the intensity changes in some subfigures, particularly (a), (c), and (i), are difficult to discern. Improving the visibility of these changes would enhance the figure's clarity.

> The unclarity of figure 3 was also criticised by reviewer 1. The intensity changes are now visible with the change of style to a pseudocolor plot.

5. Page 8, Line 188: The term "electrolyte under the electrode" should refer to the water used in the simulation, as mentioned on Page 7, Line 180. This needs to be corrected.

> This is true and was corrected accordingly.

6. Page 9, Figure 4: The manuscript should clarify whether the simulated B1 field intensity is derived from the B1 field vector as a whole or just from the component perpendicular to the B0 field.

> Thank you again for thoroughly reading through the manuscript. The according statement is added to the main text and the figure caption.

7. Page 14, Table B1: The nutation frequency listed for a 0.1 mm distance and 5 mm PEEK does not match the label of Figure B2(f). This discrepancy needs to be addressed.

> This is true and was corrected accordingly. In the course of reviewing the nutation data, the data point at time t = 0 μs with zero intensity was added to all nutation curves. This changed all numerical values of nutation frequencies and field enhancements to a small extent. The overall correlation and accordance of data was not influenced.

8. Figure B1 (a) and Figure B2 (e): Although these figures display obviously different nutation curves, they share the same nutation frequency. An explanation for this should be provided.

This due to the limited number of discrete values chosen for the nutation experiment. 80 different pulse length were tested and zero filling with a factor of 2 was used. This resulted in 160 discrete values for the nutation frequency. Apparently, these two experiments showed the highest value for the exact same nutation frequency. An according statement was added to the figure captions.

9. Figure B2: The nutation curves in (a) and (c) indicate a significant degradation in B1 field homogeneity. This degradation only occurs when the discs' thickness is 1 mm and the distance is 0.1 mm. The reasons behind this specific case should be explored.

This is due to the way of integration of peaks for the evaluation of nutation experiments. To distinguish, which resonance in the 1H spectra can be assigned to the water signal from between the coins, 1H CSI was applied before nutation experiments. Water from outside the gap between the coins, e.g. in the thin film between PEEK cylinders and the glass tube, showed a significant different chemical shift. Using CSI, this could be distinguished from one another, and the integration value for nutation experiments was chosen accordingly. In the case of figure B2 c) two resonances could not be resolved entirely, but instead another component seems to be mixed into the signal, as also a second small peak in the nutation frequency plot is apparent. The explanation of the way of integration was complemented in the text of appendix B.

10. The changes in B1 field homogeneity, as indicated by the nutation curves in figure B1 and B2, should be compared with simulation results. This comparison could provide further insights.

Since we believe these changes in B1 field homogeneity are due to the integration boundaries, as described above, this experimental data was not further compared to simulation results.

Bruce Balcom:

The paper by Schatz et al addresses a very important and interesting topic related to MR/MRI studies of electrochemical cells. On the surface the necessity of a strong static field, Bo, switched magnetic fields Gx, Gy and Gz, in addition to a radio frequency excitation field B1, would seem to be problematic in a device that must contain significant conductive and often metallic structures namely electrodes and current collectors for the electrochemical device and ancillary electrical connections. Nevertheless, remarkable progress has been made with MR/MRI studies of electrochemical cells.

Schatz and coworkers have done a good job of assembling and describing the issues. MR/MRI practioners are very enthusiastic about in operando studies but traditional battery researchers and industry are often resistant, since the batteries employed for in operando MR/MRI studies usually don't look like realistic and familiar batteries.

Resolution of this issue in the context of EM compatibility (Bo, Gx,y,z and B1) is very important for wider acceptance of MR/MRI battery in operando studies.

The authors have done a good job of assembling and assessing the pertinent literature with I fear one significant exception. On page 3 the authors assign to Zhang and Zwanziger (2011) the idea of a parallel plate resonator (PPR) for RF excitation in fuel cells. The Zhang and Zwanziger paper is however about the design of an MR compatible electrochemical cell in a 5 mm NMR tube. It is not about a parallel plate resonator, nor a fuel cell. I think the authors may have been intending to reference another work with the same last name of the first author, also in 2011, Zhang and Balcom "Magnetic Resonance Imaging", in 'PEM Fuel Cell Diagnostic Tools', Eds. Wang, H., Yuan, X. and Li, H., Taylor and Francis, Oxford, UK (2011) 229-254.

This paper summarized work by Zhang related to use of the PPR in nafion based fuel cells. The work summarized came from two JMR papers in 2008, one from J Power source 2011 and Can J Chem 2011. The later paper explicitly describes and analyzes the PPR.

The PPR resonator was designed to satisfy the EM compatibility issues outlined in the Schatz paper. I will address this further below but first I will mention that the fuel cell studies of 2008 to 2011 were more recently rejuvenated for studies of lithium ion batteries. There have been a number of published studies, but I will highlight two in particular, (i) Aguilera JMR 2021 (cover photo for the issue). This paper employed extensive use of simulation to design the PPR and critically introduced a cartridge like removable cell. The cell has electrodes that are parallel to the plates of the PPR with all of the attendant EM compatibility benefits. The cartridge is removable permitting multiplexing of samples and the 7Li battery cartridge looks like a battery to traditional battery people. (ii) Goward and coworkers have continued to use the PPR resonator and battery cartridge idea in a series of studies, notably Sanders et al Carbon (2022) wherein they showed both imaging and spectroscopic studies of a lithium ion battery. The PPR / battery cartridge was proven to be a sensitive implementation in the Sanders paper because of the significant sample volume, which is implicit in the resonator design.

But now why does the PPR / cartridge idea work and how does it help EM compatibility? It is easy to explain in words. These reasons are interwoven in the papers mentioned, but the simplest summary is probably the Zhang 2011 chapter. RF shielding must be avoided so that means B1 parallel to the electrodes and other conductive structures as Schatz and coworkers describe. If the electrochemical cell is a capacitor-like design (as in the 7Li cartridge and nafion fuel cell studies) with parallel conductive surfaces (electrodes), there are two possible orientations for Bo (parallel and perpendicular to the plates). The Bo direction will be z of course. It is advantageous for the Bo field, in terms of minimizing susceptibility distortion of the static field experienced by the sample, for the Bo field to be oriented parallel to the conducting surfaces. Okay there are two ways to be parallel to the plates but one of them must be the direction of the B1 field. It must be perpendicular to Bo and parallel to the plates. Let's identify this B1 direction as x. The Bo field, z, once again is parallel to the conductive surfaces of the battery and RF probe. In this geometry then the electrolyte will span the space between the electrodes which will be y.

There is a more subtle but very important consideration for why this geometry is advantageous. It is well known but not always fully appreciated that the Gx, Gy and Gz gradients *all* have z directed fields that vary in the chosen direction (Gx is dBo/dx with Bo z directed). Eddy currents will be at their worst when a z directed field impinges, perpendicular, to a conductive surface and gradients are switched. The emf driving eddy currents results from the time rate of change of flux *through the conducting surface*. The worst case for eddy currents will be a z directed field impinging on a planar conductor that is transverse. The PPR / cartridge combination with the conductors parallel to Bo have minimal conducting cross section in the Bo direction meaning that the emf induced is minimized, minimizing eddy currents. This is true regardless of the direction in which one wishes to do imaging. The above reasoning does not require detailed simulation, it is simple EM considerations, but it is backed up by simulation.

Schatz and coworkers employ electrodes that are perpendicular to the Bo field with gradients that are z directed. This is required given the initial geometry of their cell, magnet and gradient coil. That does not mean they can't achieve good results, they do achieve good results. It does however mean the underlying geometry is non-ideal from an EM compatibility point of view.

The above line of reasoning would be advantageous to include or at least to reference in the paper. It is easy for people to get lost in results and simulation.

On a significant but less serious note, the authors at various times state that SPI, SPRITE, constant time, and chemical shift imaging profile approaches to imaging are advantageous. The implementation that is not explicitly stated, that is similar, is spin echo SPI. The first three methods are FID based. The later two are based on spin echoes. They similar because spatial encoding is purely by phase encode gradients. This is advantageous because if there are susceptibility induced Bo distortions, pure phase encoding will not lead to geometric distortion. The distorted Bo field instead results in a local image contrast (T2* or T2) which may be removed or controlled through choice of imaging parameters. The pure phase encode approach is also robust to image distortion due to eddy currents. With frequency encode imaging eddy currents distort the image geometry. An inhomogeneous static field will also lead to image distortion in frequency encode imaging. These two ejects can make it dijicult to discriminate B1 inhomogeneity ejects, from Bo inhomogeneity ejects, from eddy current ejects with frequency encode imaging. Eddy currents manifest in pure phase encode image as a change in the image field of view, but not geometric distortion of the object geometry.

Bruce J. Balcom

MRI Research Centre, Department of Physics

University of New Brunswick

Dear Prof. Balcom,

Thank you for your valuable feedback and corrections, for thorough reading and detailed explanations.

1) You were right, that we indeed cited the wrong Zhang 2011 reference in line 77. Thank you again for indicating this mistake. We have corrected it accordingly. Also we have added a concise explanation how the cell setup presented therein minimises distortions caused by all electromagnetic fields applied (B0, B1, Gx Gy Gz).

2) It is also true that the presented cell design does not reduce eddy currents to a minimum that are caused by magnetic field gradients, as they are directed perpendicularly to the conductive surface. We added this information in line 101f. Also, we give an outlook in line 319ff how the distortions caused by magnetic field gradients could be investigated in future studies.

3) Inhomogeneities of B0 field would also lead to distortions of the frequency encoded 1H profile we presented. As we assume a well shimmed sample, we neglect this contribution to alterations in the 1H image. This was added in line 234f. Also, we added line 238f in response of your comment, that eddy currents would be manifested in purely phase encoded images as change of the image FOV.